# Complexing the Oncolytic Adenoviruses Ad∆∆ and Ad-3∆-A20T with Cationic Nanoparticles Enhances Viral Infection and Spread in Prostate and Pancreatic Cancer Models

**DOI:** 10.3390/ijms23168884

**Published:** 2022-08-10

**Authors:** Yang Kee Stella Man, Carmen Aguirre-Hernandez, Adrian Fernandez, Pilar Martin-Duque, Rebeca González-Pastor, Gunnel Halldén

**Affiliations:** 1Centre for Biomarkers and Biotherapeutics, Barts Cancer Institute, Queen Mary University of London, London EC1M 6BQ, UK; 2Centre for Sarcoma Molecular Pathology, Institute of Cancer Research (ICR), Sutton SM2 5NG, UK; 3Department of Surgery and Cancer, Imperial College London, London W12 0HS, UK; 4Centro Nacional de Investigaciones Oncológicas (CNIO), 28029 Madrid, Spain; 5Instituto Aragonés de Ciencias de la Salud/IIS Aragón, 50009 Zaragoza, Spain; 6Fundación Araid, 50018 Zaragoza, Spain; 7Departamento de Cirugía, Facultad de Medicina, Universidad de Zaragoza, 50009 Zaragoza, Spain; 8Centro de Investigación Biomédica (CENBIO), Facultad de Ciencias de la Salud Eugenio Espejo, Universidad UTE, Quito 170527, Ecuador

**Keywords:** oncolytic adenovirus, tumour-selective, AuNP, PCa, PDAC

## Abstract

Oncolytic adenoviruses (OAd) can be employed to efficiently eliminate cancer cells through multiple mechanisms of action including cell lysis and immune activation. Our OAds, AdΔΔ and Ad-3∆-A20T, selectively infect, replicate in, and kill adenocarcinoma cells with the added benefit of re-sensitising drug-resistant cells in preclinical models. Further modifications are required to enable systemic delivery in patients due to the rapid hepatic elimination and neutralisation by blood factors and antibodies. Here, we show data that support the use of coating OAds with gold nanoparticles (AuNPs) as a possible new method of virus modification to help augment tumour uptake. The pre-incubation of cationic AuNPs with AdΔΔ, Ad-3∆-A20T and wild type adenovirus (Ad5wt) was performed prior to infection of prostate/pancreatic cancer cell lines (22Rv, PC3, Panc04.03, PT45) and a pancreatic stellate cell line (PS1). Levels of viral infection, replication and cell viability were quantified 24–72 h post-infection in the presence and absence of AuNPs. Viral spread was assessed in organotypic cultures. The presence of AuNPs significantly increased the uptake of Ad∆∆, Ad-3∆-A20T and Ad5wt in all the cell lines tested (ranging from 1.5-fold to 40-fold), compared to virus alone, with the greatest uptake observed in PS1, a usually adenovirus-resistant cell line. Pre-coating the AdΔΔ and Ad-3∆-A20T with AuNPs also increased viral replication, leading to enhanced cell killing, with maximal effect in the most virus-insensitive cells (from 1.4-fold to 5-fold). To conclude, the electrostatic association of virus with cationic agents provides a new avenue to increase the dose in tumour lesions and potentially protect the virus from detrimental blood factor binding. Such an approach warrants further investigation for clinical translation.

## 1. Introduction

Prostate cancer (PCa) is the second most common cancer diagnosed in men, with up to 4% (375,000/year) of cancer-related deaths globally [1]. Pancreatic ductal adenocarcinoma (PDAC) is less common but is more lethal (over 430,000/year) in both men and women [2]. The 5-year survival rate for PDAC patients is dismal at only 9%, while the corresponding survival rate for PCa is 85%. In the late stages of both cancer types, treatment-resistance to all current therapeutics inevitably arises, accompanied by severe morbidity [3,4]. Therefore, there is an immediate need to develop novel therapies that negate drug-resistance. Employing engineered oncolytic adenoviral (OAd) mutants that selectively replicate in and eliminate cancer cells, leaving normal cells unharmed, is a promising therapeutic strategy. Powerful and specific cancer cell killing by OAds via cell lysis releases novel tumour antigens and pathogen-associated molecular patterns (PAMPs). These and damage-associated molecular patterns (DAMPS) promote activation of the host anti-tumour immune responses and abscopal effects [5,6,7]. For example, when OAds were combined with immune checkpoint inhibitors (e.g., anti-PD-1) or cytokines, the activation of the host immune system resulted in the elimination of non-treated tumours and long-term immunity [8,9]. Another example of the abscopal effect is the spread of OAds from the tumour site to distant tumours via extracellular vesicles [10]. Various engineered OAd mutants have been evaluated in clinical trials including both PCa and PDAC patients [11,12,13,14,15,16,17]. Although safety was unanimously demonstrated and local tumour regression was reported after intra-tumoural administration, complete tumour elimination remains problematic. The main obstacles include poor viral uptake and spread within the tumour microenvironment (TME) and an insufficient number of viral particles reaching distant metastatic lesions after systemic administration [18,19].

We previously engineered an oncolytic adenovirus, AdΔΔ, to exploit the typical alterations in late-stage PCa and PDAC cancers, which include the deregulation of cell cycle control by the pRb/p16-p53 pathways and apoptosis mechanisms [20,21,22,23]. The selectivity and potency of AdΔΔ are achieved by deletions of the viral E1ACR2-domain (pRb-binding) and the anti-apoptotic Bcl2-homologue E1B19K [21,24,25,26]. As a result, AdΔΔ replicates only in cells with deregulated cell cycle pathways and promotes the induction of apoptosis in response to cytotoxic drugs even in drug-resistant tumours [27]. AdΔΔ is highly efficacious, selective and non-toxic in preclinical models of PCa and PDAC including cultured cells and human tumour xenografts in athymic mice [20,26]. However, to eliminate metastatic lesions in patients, the systemic delivery of OAds would be necessary, which is not currently feasible due to rapid hepatic elimination and high affinity binding to erythrocytes [18,19,28]. To improve the bioavailability of the virus in vivo, we genetically engineered a retargeted version of Ad∆∆, the Ad-3∆-A20T mutant. In addition to the deletions in AdΔΔ, Ad-3∆-A20T has a third deletion of the immune-regulatory E3gp19K protein and altered receptor tropism [29]. More specifically, the native binding to the Coxsackievirus and Adenovirus Receptor (CAR) is ablated by deletion of the TAYT-motif in the fibre knob region of the virus, preventing erythrocyte-binding, and by insertion of a 20-amino acid peptide from the Foot-and-Mouth-Disease-Virus (FMDV). The mutant retargets infection to αvβ6-integrin, a cell surface receptor exclusively expressed in solid epithelial cancers and at high levels in PDAC cells [29,30,31]. The depletion of E3gp19K re-activates the host immune responses to infected cancer cells by enabling efficient MHC-antigen presentation [32]. The Ad-3∆-A20T mutant is highly selective and potent, infecting and propagating in PDAC cells after systemic delivery in vivo [33].

We previously reported that positively charged gold nanoparticles (AuNP) could coat the highly negatively charged viral particle (−6 mV; [34]) through electrostatic interactions, resulting in more efficient infection and propagation of virus in cancer cells [35,36]. We hypothesized that coating Ad∆∆ and Ad-3∆-A20T with the cationic AuNPs would shield the viruses from hepatic Kupffer cells and erythrocytes, increasing the circulating half-life and local tumour concentration compared to their uncoated counterparts. Here, we demonstrate that association of the cationic AuNPs with Ad5wt (wild-type Adenovirus serotype 5), Ad∆∆ and Ad-3∆-A20T did not impede viral functions but increased viral efficacy in several models of PCa and PDAC tumours via increased cellular uptake, replication and specific cancer cell killing. These data indicate that complexing the viral particle with cationic molecules or agents enables a practical and versatile method for augmenting the efficacy of the virus, rather than introducing more genetic modifications that could potentially be detrimental for the virus.

## 2. Results

### 2.1. Gold Nanoparticles (AuNP) Enhance Adenovirus Uptake in PCa and PDAC Adenocarcinoma Cells

We previously demonstrated that AdΔΔ potently enhanced mitoxantrone-induced cell killing in the PC3 prostate cancer cell line, despite the poor sensitivity to Ad-infection in these cells [20,24,26]. To investigate if viral uptake could be increased in PC3 cells, OAds were incubated with spherical, 14 nm cationic AuNPs functionalized with ammonium groups prior to infection to allow binding of the viral coat proteins by electrostatic interactions at non-cytotoxic AuNP concentrations of up to 0.5 pmol, as determined in previous studies [35]. PC3 cells infected with the replicating EGFP-expressing Ad5wtGFP mutant alone at 10, 100 and 500 ppc resulted in low levels of viral EGFP-expression, while viral pre-incubation with AuNPs showed significantly increased viral uptake at all tested doses (Figure 1A). The uptake of Ad5wtGFP, determined as EGFP-expression, increased in a dose-dependent manner, from 0.2 ± 0.1% to 14 ± 9% (10 ppc); from 2 ± 1% to 62 ± 17% (100 ppc); and from 7 ± 3% to 85 ± 9% (500 ppc) of cells. The more Ad-sensitive 22Rv cells also showed significant increases in viral uptake in the presence of AuNPs from 13 ± 7% to 29 ± 13% (10 ppc) and from 41 ± 13% to 63 ± 18% (50 ppc) of cells (Figure 1A). To further explore whether the AuNPs promoted uptake in other cell types, the PDAC, Panc04.03 and PT45 cells were infected with Ad5wtGFP ± AuNPs under similar conditions (Figure 1B). A significant time-dependent increase in viral EGFP-expression/uptake was noted in both cell lines with the greatest increases after 48 h, from 3 ± 0.3% to 58 ± 3% (100 ppc; Panc04.03) and from 4 ± 0.1% to 49 ± 6% (100 ppc; PT45).

A major obstacle for viral uptake and spread in PDAC *in situ* is the dense stroma surrounding the tumour mass that is mainly comprised of cancer-associated stellate fibroblasts [37]. When the Ad-resistant pancreatic stellate cell line PS1 was infected with Ad5wtGFP ± AuNPs, a highly significant increase in uptake and EGFP-expression was observed after 24 and 48 h, from 0.2% to 26 ± 8% and from 1% to 50 ± 9%, (500 ppc), respectively. The overall increase in the fluorescent EGFP intensity was due to the greater number of infected cells, and the increased viral uptake per cell in both PCa and PDAC models is evident from the microscopic images (Figure 1C).

### 2.2. The AuNP-Dependent Increased Viral Uptake in PCa and PDAC Cells Results in Enhanced Ad∆∆-Induced Cell Killing

To examine if the increased viral uptake was paralleled by increased cell killing, PC3 and 22Rv cells were infected with increasing doses with and without AuNPs. In the PC3 cells, a five-fold decrease in EC_50_ values was observed following AdΔΔ/AuNPs infection compared to Ad∆∆ alone, from 3119 ± 785 ppc to 625 ± 211 ppc (Figure 2A,B). In the Ad-sensitive 22Rv cells, a lower but significant three-fold decrease in EC_50_-values was recorded, from 126 ± 8 to 46 ± 9 ppc (Figure 2A,B). These findings demonstrate that the enhancement of cell killing is attributable to the increased uptake of Ad∆∆-virus in the presence of non-toxic doses of AuNPs.

Mitoxantrone is frequently administered to patients with late stage PCa. To assess whether increases in cytotoxicity mediated by AuNP/AdΔΔ were maintained in the presence of mitoxantrone, the PC3 cells were treated with AdΔΔ at 500 and 750 ppc alone, or in combination with 450 nM mitoxantrone, in the presence or absence of AuNPs. AdΔΔ/AuNP infection resulted in increased cell killing both alone and in combination with mitoxantrone, compared to AdΔΔ alone (Figure 2C). AuNP-coating increased AdΔΔ (500 ppc) cell killing from 16 ± 5% to 70 ± 9% (no mitoxantrone) and from 63 ± 7% to 91 ± 3% (with mitoxantrone). Similar increases were also observed at the higher dose of 750 ppc. The presence of AuNPs in combination with mitoxantrone did not affect the cell death response compared to mitoxantrone alone, 31 ± 1% (mitoxantrone alone) and 27 ± 3% (mitoxantrone + AuNP) (Figure 2C), supporting the notion that AuNP exerts its effects via interaction with the virus and not with mitoxantrone. It is notable that AuNPs alone did not affect cell viability at the selected doses in any of the cell lines tested, as demonstrated in PC3 and 22Rv cells (Mock in Figure 2C and Appendix A).

Ad∆∆-mediated cell killing was also significantly improved in the PDAC and PS1 cell lines (Figure 2D). The EC_50_ values in Panc04.03 cells decreased from 262 ± 25 ppc (Ad∆∆) to 98 ± 10 ppc (Ad∆∆/AuNP), and in PT45, from 126 ± 12 ppc (Ad∆∆) to 56 ± 4 ppc (Ad∆∆/AuNP). Significantly enhanced cell killing was also observed in the Ad-resistant PS1 cells, whereby EC_50_ values reduced from 2300 ± 150 ppc (Ad∆∆) to 1600 ± 100 ppc (Ad∆∆/AuNP) (Figure 2D). 

### 2.3. Complexing of Ad∆∆ with AuNPs Increases Ad∆∆ Replication in All Tested Cell Lines

To determine whether the higher levels of cell killing caused by the potent expression of early viral genes is a consequence of the increased viral uptake, changes in viral replication were examined (Figure 3). In the PC3 cells, the presence of AuNPs significantly increased AdΔΔ replication compared to AdΔΔ alone, from 1.5 × 10^7^ ± 0.5 × 10^7^ pfu/mL to 7 × 10^8^ ± 1 × 10^8^ pfu/mL (48 h), and 3 × 10^7^ ± 0.5 × 10^7^ pfu/mL to 6.9 × 10^8^ ± 1 × 10^8^ pfu/mL (72 h) (Figure 3A, right). In the 22Rv cells, Ad∆∆ replication increased from 9 × 10^6^ ± 0.1 × 10^5^ pfu/mL to 3 × 10^7^ ± 0.5 × 10^7^ pfu/mL in the presence of AuNPs after 48 h, with no further significant increases after 72 h (Figure 3A, left). In the PC3 cells, an increase in virus replication was also observed in the presence of AuNP when Ad∆∆ and mitoxantrone were added simultaneously, from 7 × 10^5^ ± 0.5 × 10^5^ to 5 × 10^6^ ± 1 × 10^6^ pfu/mL (Figure 3A, right). However, drug-mediated attenuation of virus replication remained as compared to the cells that were infected with virus alone, which is in agreement with our previous studies [24,26]: 7 × 10^5^ ± 0.5 × 10^5^ (with mitoxantrone at 48 h) vs. 1.5 × 10^7^± 0.5 × 10^7^ pfu/mL (without mitoxantrone at 48 h) and 5 × 10^6^ ± 1 × 10^6^ pfu/mL (mitoxantrone + AuNP at 48 h). Although mitoxantrone attenuated overall viral replication, the presence of AuNPs was able to partially rescue this effect via an increase in viral uptake.

To assess whether the AuNPs supported viral infection in vivo, immunodeficient animals with PC3 xenografts were intravenously treated with Ad∆∆ ± AuNPs. At 25 days post-administration, E1A expression was detected at high levels in xenografts obtained from both treatment groups, although, under these conditions, significant differences could not be determined (Appendix A).

Ad∆∆ replication in the PDAC cells, Panc04.03 and PT45, was markedly increased in the presence of AuNPs, as also observed in PCa cells, from 5 × 10^8^ to 2 × 10^10^ pfu/mL (Panc04.03) and from 7 × 10^7^ to 2 × 10^10^ pfu/mL (PT45) (Figure 3B). The PS1 stellate cells supported only low levels of Ad∆∆ replication when infected with virus alone (1 × 10^6^ ± 0.2 × 10^5^ pfu/mL), but in the presence of AuNP, Ad∆∆ replication significantly increased (7 × 10^9^ ± 1 × 10^8^ pfu/mL).

Taken together, these findings demonstrate that Ad∆∆ efficiently infected cells when complexed with the AuNPs both in vitro and in vivo, and the higher levels of viral uptake resulted in increased replication and cell killing.

### 2.4. Efficient Elimination of Panc04.03 Cells in Organotypic Co-Cultures with PS1 Cells after Infection with Ad5wt and Ad∆∆ Precoated with AuNPs

To explore whether the enhanced cell killing in the presence of AuNPs could also eliminate cancer cells in a more physiologically relevant model system, the Panc04.03 cells were co-cultured with PS1 cells in three-dimensional organotypic collagen-Matrigel cultures and infected with Ad5wt and Ad∆∆ in the presence or absence of AuNPs. Potent dose-dependent (1000 and 2000 ppc) cell killing was observed with Ad5wt alone and was enhanced in combination with AuNPs at 5 days post-infection (Figure 3C). High levels of E1A were detected in epithelial cell layers of the organotypic cultures after treatment with Ad5wt/AuNPs (2000 ppc), and to a lesser degree with a lower dose (1000 ppc) (Figure 3D, green staining). Less E1A expression was detected in cells infected with Ad5wt alone compared to Ad5wt/AuNPs at both doses. In cultures infected with the more potent Ad∆∆ at a low dose (50 ppc), the efficient elimination of all the infected cells (both in presence and absence of AuNP) was observed at 1- and 6-days post-treatment (Appendix A). Overall, both Ad5wt and Ad∆∆ efficiently infected and lysed PDAC cells cocultured with stromal (PS1) cells in a three-dimensional matrix, even in the presence of AuNPs, suggesting efficient viral propagation and spread.

### 2.5. Infectivity and Replication of the αvß6-Integrin Targeted Viral Mutant Ad-3∆-A20T Is Improved in Combination with AuNPs

To assess whether the enhanced infection was specific for the Ad5wt and Ad∆∆ mutant that have identical viral coats, uptake of the retargeted Ad-3∆-A20T mutant was investigated. Ad-3∆-A20T has the FMDV peptide inserted in the fibre knob domain that contributes an additional positive charge to the viral coat, but the overall charge of Ad-3∆-A20T remains negative. We previously demonstrated high levels of αvß6-integrin expression in PDAC cells and selected these cells for further studies with Ad-3∆-A20T and its GFP expressing variant, Ad-3∆-A20T-GFP (EGFP inserted in the E3gp19K region) [29]. In Panc04.03 cells, viral GFP-expression/uptake increased from 2.5 ± 0.5 to 13.4 ± 2.0% (24 h; five-fold) and from 4.4 ± 0.5 to 32.5 ± 6% (48 h; seven-fold) in the presence of AuNPs (Figure 4A). In PT45 cells, viral uptake increased to a lower extent, from 3.0 ± 2 to 18.2 ± 7% (48 h) (Figure 4A). The most significant increase was observed in the PS1 cells that was 15-fold higher in the presence of AuNPs, from 2.2 ± 3.5 to 33.4 ± 8% (48 h), also shown in representative fluorescent micrographs (Figure 4A, lower panel).

To verify the viral replication of Ad-3∆-A20T in the presence of AuNP, replication rates were determined over time (24–72 h), relative to Ad∆∆ replication ± AuNPs (relative genome amplification, Figure 4B). The addition of nanoparticles significantly increased viral genome amplification over time for both Ad∆∆ and Ad-3∆-A20T in all three cell lines. The most pronounced increase was observed in the poorly infectible PS1 cells by up to 1000-fold for Ad-3∆-A20T/AuNPs at 24 h post-infection, and this continued to escalate during the entire time-course (Figure 4B, lower right panel). Similarly, a significant increase was also observed in Ad∆∆/AuNP infected PS1 cells, by up to 8000-fold at 24 h post-infection that remained elevated up to 72 h (Figure 4B, lower left panel). In PT45 cells, both viruses showed higher levels (70-100-fold) of genome amplification at 24 h after infection in the presence of AuNPs with the same fold-change from 24–72 h (Figure 4B, middle panels). A noticeable increase was also observed in Panc04.03 cells infected with Ad∆∆/AuNP (60-fold, at 24 h) and to a lower degree after Ad-3∆-A20T/AuNP infection (six-fold, at 24 h) (Figure 4B, upper panels). Overall, the viral genome amplification in all three cell lines was elevated in the presence of AuNPs for both viruses, although amplification rates ± AuNPs remain relatively similar, as evident by the parallel curves on the time-response graphs.

The viral spread of Ad-3∆-A20T/AuNP was similar to or greater than that observed with virus alone in three-dimensional co-cultures of Panc04.03 and PS1 cells at a low dose (10 ppc) (Appendix A). A trend towards cell detachment was observed 2 days after infection with both Ad-3∆-A20T or Ad∆∆.

## 3. Discussion

The native tropism of adenovirus to epithelial cells makes adenocarcinomas the ideal targets for OAds. Potent tumour elimination without toxic side effects has been reported from clinical trials with various OAds administered locally to patients with solid cancers including PCa and PDAC [11,12,13,14,15,16,17]. However, the elimination of distant metastatic lesions has not been successful in most clinical trials. The high affinity binding of OAds to CAR on erythrocytes, the neutralisation of adenovirus by circulating antibodies and binding to blood factors such as Factor X (FX) that mediate the rapid hepatic elimination of virus within minutes, explain the lack of curative effects by systemic delivery [19,38,39,40]. Taken together, insufficient doses of OAds reach the metastatic cells. Additionally, late-stage tumour cells frequently dedifferentiate and express low levels of viral receptors that in turn prevent efficient uptake. Once tumour cells are infected, viral replication and spread are typically prevented particularly in PDACs, due to the surrounding dense stromal fibroblasts that form an impenetrable physical barrier to virus and immune cells [37]. Thus, to further improve OAd efficacy, there is an important need to increase the viral dose at metastatic lesions. Here, we demonstrate that coating the virion with positively charged gold nanoparticles (AuNPs) significantly increases cellular uptake and promotes enhanced levels of viral replication and spread in both two- and three-dimensional co-cultures of PCa and PDAC models.

The adenovirus coat is composed of highly negatively-charged proteins, generating an anionic particle [37] that favours the electrostatic binding of the positively charged AuNPs to form an overall positively charged particle. The neutralisation of the viral negative surface charges likely facilitates the attachment of virus to negatively charged cell membranes by bringing the viral fibre and penton proteins closer to the cellular CAR and integrin receptors. In this study, we demonstrated that OAds complexed with AuNPs significantly enhanced adenoviral infection in both PCa and PDAC cell lines, as well as in the transformed pancreatic PS1 stellate cells. The greatest effects were observed in the PC3 and PS1 cells that are highly insensitive, to adenoviral infection with up to 70-fold increases in uptake compared to naked virus. Interestingly, the epithelial PC3 cells are the most adenovirus-resistant adenocarcinoma cell line, requiring higher doses of >100-fold to achieve the same level of infection than 22Rv cells and other PCa cell lines [23,26]. It has been well established that mesenchymal cells—such as pancreatic stellate cells, as modelled by the cell line PS1—are not infectible or are poorly infectible, requiring higher doses by up 1000-fold than the epithelial PDAC cell lines, suggesting that viral uptake mainly depends on non-receptor mediated mechanisms in these cells [29]. Strikingly, by coating Ad5wtGFP with AuNPs, 55% of PS1 was infected with the virus, compared to undetectable levels with virus alone 48 h after infection. Both PC3 and PS1 cells have low levels of cellular uptake receptors such as CAR, α_v_β3-, α_v_β5- and α_v_β6-integrins, partly explaining the low levels of infection with virus alone [26,29]. Based on observations in the more virus-sensitive 22Rv and PDAC cell lines, efficient receptor-mediated viral uptake was further enhanced after coating Ads with AuNPs. Complexing Ad5wtGFP, Ad∆∆ and Ad-3∆-A20T with the AuNPs significantly enhanced viral uptake in all cell lines tested. Taken together, we suggest that in addition to integrin-mediated endocytosis, non-receptor mediated uptake mechanisms also contribute to enhanced infection in all cell lines via cationic interactions with the cell membrane.

Importantly, the increases in viral uptake were reflected in the greatly enhanced viral replication and cell killing with all three viruses in all cell lines tested, with the greatest effects in the most virus-insensitive cells (PC3 and PS1). When PC3 cells were infected with Ad∆∆ in the presence of mitoxantrone, viral replication was significantly reduced but was partially restored in the presence of AuNPs. Both cytotoxicity and replication were maintained at higher levels, despite the inhibitory effects on viral propagation by mitoxantrone. This was most likely due to the increased number of viral particles entering the cells in the presence of AuNPs, enabling more viral genomes to be amplified and replicated, albeit at lower levels than in cells without drug treatment [24,26].

Multiple strategies have previously been explored to improve the cellular uptake of OAds and to prevent the rapid hepatic elimination after systemic delivery, including shielding of the viral particles with cationic agents. For example, the incorporation of a poly-Lysine sequence in the fibre knob to generate CRAd-S-pk7 enhanced tumour cell uptake, although, intra-tumoural spread was limited in preclinical glioma models [41,42]. The delivery of virus using cell carriers to act as a ‘Trojan horse’, e.g., a decoy strategy to deliver the OAds to specific tumour sites by infecting autologous or allogeneic mesenchymal stem cells, has successfully been tested. For example, CRAd-S-pk7 was effectively delivered to glioma xenografts in murine in vivo models using an immortalised human neural stem cell line [43], an approach that was later used in a Phase I clinical trial (NCT03072134). Other studies used autologous mesenchymal stem cells as virus carriers, e.g., virally transduced stem cells derived from adipose tissue were successfully delivered to ovarian tumours in a Phase I trial [44]. E1-modified mesenchymal stem cells used as a carrier and packaging system for Ads were developed for evaluation in preclinical models of PCa [45]. Human immortalised stem cells expressing E1A/B effectively produced a non-replicating virus that was released and infected the murine TRAMPC PCa tumours in vivo. Other strategies for the systemic delivery of virus include the transduction of extracellular vesicles from bone marrow mesenchymal stem cells [46] or the isolation of vesicles from infected cultured cancer cells (HCT116) [10]. There are no reports of any toxicity in association with using cell carrier or extracellular vesicle systems in vivo or in clinical studies. However, difficulties with isolating stem cells or extracellular vesicles from patients, the low infection-efficiency of virus ex vivo and the sufficient expansion of virus without losing their targeting potential are some of the many issues that need addressing to achieve clinical efficacy [47]. Each strategy has advantages and disadvantages, and while the cell/extracellular vesicle carrier approach clearly protects OAds from blood factor binding and enhances viral uptake in tumour cells, penetrating the dense stromal compartment—especially in PDAC—remains an obstacle for efficient spread within the tumour. Our AuNP-coated OAds are smaller, easier to produce, more versatile than cell therapies and can penetrate and spread within the tumour microenvironment. There are currently a variety of methods that allow convenient surface modification and functionalization of AuNPs to target specific tumour antigens for additional tumour selectivity [48]. In contrast to the versatility provided by chemical modifications, the genetic modification of Ad vectors is limited by factors such as the structural instability and impairment of vector production [49]. Unlike our tropism-modified FMDV-targeting Ad-3∆-A20T, AuNP coating allows a more versatile cell line-independent enhancement of OAds infection. In addition, the greater increase in initial viral uptake that is less dependent on receptor levels would ensure potent viral replication and spread following the infection of tumour cells. Although gold particles may accumulate in the liver to toxic levels, we previously demonstrated insignificant hepatotoxicity and no significant differences in the number of white blood cells when Ad5/AuNP complexes or AuNPs alone were administered systemically in murine models at high doses [36]. Further development of other non-toxic cationic compounds as alternative OAd-shielding agents could support clinical translation.

Our findings are in agreement with previous reports that complexing cationic agents with OAds benefits viral uptake and selective replication in tumour cells. In the event that viral uptake is also increased in healthy cells, replication and spread cannot proceed due to gene deletions in the OAds. We demonstrated that cationic nanoparticles enhanced viral uptake and, therefore, the potency of OAds in preclinical studies. More extensive and in depth in vivo studies are required to develop future clinical applications. However, our preclinical data suggest that shielding the virus with cationic agents may protect it from premature inactivation by detrimental blood factor-binding when delivered via the blood stream. We anticipate that AuNPs could be replaced by biodegradable compounds to avoid potential cytotoxicity in patients. To this end, we are currently exploring alternative physiological strategies to protect the virion and enable the incorporation of tumour-targeting ligands for specific uptake with cancer cell-selective virus replication. Our preliminary findings suggest that biodegradable cationic amino acids and proteins are feasible alternatives to AuNPs to safely and efficaciously deliver OAds in the blood stream.

## 4. Materials and Methods

### 4.1. Cell Lines and Culture Conditions

Human prostate cancer PC3 and 22Rv1 (ATCC, LGC Standards, UK), and human pancreatic ductal adenocarcinoma (PDAC) Panc04.03 (ATCC) and PT45 (Prof H. Kalthoff, Comprehensive Cancer Centre, Campus Kiel, Kiel, Germany) cell lines were used in the study. The PS1 cells were hTERT-immortalised human pancreatic stellate cells and were a kind gift from Prof. H. Kocher (BCI, QMUL, London, UK). The human embryonic kidney cells HEK293 and JH293 (Cell Services, Cancer Research UK, London, UK) were used for viral production and activity determination, as previously described [20]. The cells were grown at 37 °C/5% CO_2_ in Dulbecco Modified Eagle’s medium (DMEM), supplemented with 10% Fetal Bovine Serum (FBS) and 1% penicillin and streptomycin (Penicillin 10,000 units/mL, Streptomycin 10 mg/mL; P/S) (Sigma-Aldrich, St Louis, MO, USA). STR-profiling (LGC Standards, UK and Cancer Research UK, London, UK) verified the identity of stocks and all cells were mycoplasma free.

### 4.2. Viruses

The engineering and production of the viruses have previously been described [20,21,29]. Briefly, all mutants were derived from species C wild-type adenovirus type 5 (Ad5), including the modified enhanced green fluorescent protein (EGFP)-expressing mutants that express EGFP in response to E1A-expression immediately after viral uptake (Ad5wtGFP, Ad-3∆-A20T-GFP; EGFP in E3gp19K domain). The generation of Ad∆∆ (with E1ACR2 and E1B19K deletions) has previously been described [20]. The Ad-3∆-A20T is a modified version of Ad∆∆ with E3gp19K deleted and retargeted to avß6-integrins by the insertion of a 20-amino acid peptide in the fibre knob, in addition to the deletion of a TAYT-motif to ablate CAR-binding [29]. The viruses were produced, purified and characterised according to standard protocols [20,21]. The viral particle (vp) to infectious units (plaque-forming units; pfu) was 10–100 vp/pfu for all viruses, with the highest ratios obtained for FMDV-expressing mutants. All infections were performed in serum-free DMEM for 2 h/37 °C before replacement with 10% FBS/1% P/S DMEM.

### 4.3. Gold Nanoparticles (AuNP) and Ad-AuNPs Complexes

The synthesis and characterization of the gold nanoparticles (AuNP) have already been described [35,36], and a part of the AuNPs were kindly synthesized by Prof. Jesus Martinez de la Fuente (INA, Zaragoza, Spain). AuNPs consist of a gold core of 14 nm functionalized with polyethylene glycol-containing ammonium groups (PEG-NH_4_^+^) to generate positively charged particles (+30 mV). The AuNPs were stored at 4 °C in dark glass bottles to prevent non-specific binding to plastic, and aliquots were diluted immediately prior to pre-incubation with virus. Ad/AuNPs complex formation (150–280 nm diameter) were freshly prepared by the addition of AuNPs to Ad in serum-free medium (SFM) with incubation for 20 min at room temperature. The morphology and size of the AuNPs and the complexes were characterized using a Cary 50 UV-Vis spectrophotometer (Varian Inc., Agilent Technologies) by agarose gel electrophoresis and by transmission electron microscopy, FEI TECNAI T20 system at 80 kV. The AuNPs were quality checked by ensuring the presence of the expected UV-Vis spectroscopy absorption spectrum (a sharp peak at 520 nm), and a peak shift to 547 nm of Localised Surface Plasmon Resonance (LSPR) bands for Ad-AuNPs@NR4 red complexes.

### 4.4. Flow Cytometry

The cells were seeded (1 × 10^5^ cells/well) in 6-well plates 24 h prior to infection with the EGFP-expressing mutants at 10–500 ppc in serum free DMEM. Two hours later, the media were replaced with 10% FBS/DMEM. Virus and AuNP were prepared as detailed above (in Section 4.3) at the indicated ratios. Viral uptake was quantified by flow cytometry analysis (FACs) using EGFP expression as a marker, 24–72 h post-infection. The cells were detached with trypsin/EDTA and combined with non-attached cells in the media before resuspending in cold FACS buffer (0.1% BSA/DMEM). Fluorescence was detected by the FACSCalibur instrument acquiring 10,000 events per sample and analysed using the FlowJo software 8.8.6 (Tree Star Inc., Ashland, OR, USA).

### 4.5. Cell Viability Assays

The cells (1 × 10^4^ cells/well in 96-well plates) were infected with viral mutants at increasing doses in 2% FBS/1% P/S DMEM and the cell viability was quantified by the 3-(4,5-dimethylthiazol-2-yl)-5-(3-carboxymethoxyphenyl)-2-(4-sulfophenyl)-2H-tetrazolium assay (MTS; Promega, Southampton, UK) as an indirect measurement of cell death. Dose–response curves were generated to determine the virus concentration at which 50% of cells are killed (EC_50_), using untreated cells as controls at 72 h or 6d post-infection. Each data point was generated from triplicate samples and the experiments were repeated at least three times, as previously described [24,27]. To complex virus with AuNPs, each stock dilution of virus was incubated with 0.1 pmol of AuNP, such that the final concentration of AuNP per treatment was 0.03 pmol. Virus and AuNP were pre-incubated for 15 min at 24 °C prior to treatment.

### 4.6. Viral Replication Assay by TCID_50_

The cells (1 × 10^5^/6-well plate) were seeded in 10% FBS/DMEM 24 h prior to adenoviral infection when the media were replaced and virus added in serum-free media for 2 h. AuNPs (0.5 pmol) were pre-incubated with virus (10–500 ppc) for 15 min in serum-free media at 24 °C prior to treatment. After 2 h, the media were replaced with 10% FBS/DMEM and returned to the 37 °C incubator. Cells and media were collected 24–72 h post-infection, freeze-thawed three times prior to assay by TCID_50_ on JH293 cells and functional virions quantified, as previously described [24,29].

### 4.7. Viral Genome Amplification by qPCR

The ells were infected (as described above for TCID_50_) with the respective virus and harvested at 24–72 h post-infection. The cell suspensions were pelleted, snap-frozen and stored at −80 °C. DNA was extracted using the QIAamp DNA Blood Mini Kit, according to the manufacturer’s instructions (Qiagen, Hilden, Germany), and used for a quantitative PCR (qPCR) analysis, as previously described [24,29].

### 4.8. Organotypic 3-Dimensional (3D) Co-Culture Models

Three-dimensional co-culture models (organotypic cultures) consisting of cancer and stromal cells have previously been described [29]. Briefly, collagen type I was mixed with Matrigel (3:1) and allowed to solidify at 37 °C in Transwells^®^ (Corning; Sigma-Aldrich, St Louis, MO, USA) before placing into 10%FBS/DMEM in the lower chamber of 24-well plates. Panc04.03 cells were seeded together with PS1 cells (1:2; total 1 × 10^5^ cells/well) on top of the gels. After 72 h, the media in the top-chamber were replaced with serum-free DMEM and, 3 days later, viruses ± AuNPs were added at different doses and ratios. The cultures were fixed in formalin 8 days after seeding.

### 4.9. Immunohistochemistry

Organotypic cultures were formalin-fixed, paraffin embedded, dewaxed, and rehydrated in decreasing ethanol concentrations, water and PBS before sectioning onto microscope slides at 10 µm thickness. The sections were stained with Haematoxylin and Eosin (H/E) and analysed by light microscopy. Antigen retrieval was in 10 mM Sodium Citrate Buffer pH6 for 8 min, followed by water and PBS at 24 °C, and cells were permeabilised in 0.2% Triton-X/PBS in blocking buffer (2% BSA/10% FBS/PBS) for 5 mins. The sections were incubated overnight with E1A antibody (1:500; M58, GeneTex, Irvine, CA, USA) at 4 °C, followed by incubation with fluorescent-labelled secondary antibodies for 1 h (anti-mouse 488 Alexaflour 1:500, Thermo Fisher Scientific, Waltham, MA, USA) and stained with 4′,6′-diamidino-2-phenylindole (DAPI, 1 mg/mL; ThermoFisher) before mounting with FluorSave Reagent (Calbiochem, La Jolla, CA, USA). The samples were stored at −20 °C and analysed by confocal microscopy (Zeiss 710, Jena, Germany).

### 4.10. In Vivo Tumour Growth

The PC3 cells (5 × 10^6^ cells) in Matrigel (1:1) were inoculated subcutaneously in one flank of CD*nu/nu* athymic mice (Charles River, Harlow, UK). The treatments were initiated when the tumour volumes reached 80 ± 20 μL. The intravenous delivery of Ad∆∆ at 1 × 10^10^ vp/injection or Ad∆∆ pre-incubated with AuNPs for 15 min at 24 °C was performed on day 1, 3 and 5. Tumour growth, progression and animal weight were followed until the tumours reached 1.2 cm^2^ (according to UK Home Office Regulations). Each treatment group included 3 animals. The tumours were harvested 25 d after virus administration and fixed in 4% formaldehyde, sectioned and processed for histopathology with H/E and for immunohistochemistry (IHC) by staining for E1A (1:500; M58, GeneTex, Irvine, CA, USA), and detected by an HRP-conjugated secondary antibody (Dako, Copenhagen, Denmark).

## 5. Conclusions

We have demonstrated that complexing the highly anionic adenoviral particle with large cationic molecules composed of an inorganic core (Au) with covalently linked positively charged polyethylene glycol (PEG-NH_4_^+^) chains significantly increased viral uptake, replication and cell killing in PCa and PDAC in vitro models. These findings are in agreement with our previous report that describes how complexation of AuNPs with adenovirus enhanced the ionic interactions with anionic Heparin Sulfate Glycolsylationglycans (HSGAGs), which are abundantly expressed on mammalian cell membranes [35]. We demonstrate enhanced viral propagation in PCa and PDAC cells after complexing all the tested viruses (Ad5wt, Ad∆∆ and Ad-3∆-A20T) with AuNPs. Interestingly, in cells that are normally resistant to adenovirus infection, such as the PS1 cell line, significant increases in viral uptake and propagation were observed with both Ad∆∆ and Ad-3∆-A20T. Electrostatic interactions between the viral particle and the AuNPs may have generated a cationic shield that facilitated both non-receptor- and receptor-mediated uptake by bringing the virus within close proximity to the cell membrane, increasing access to extracellular receptors and promoting internalisation.

To conclude, our findings suggest that shielding OAds with cationic compounds may protect the virus from blood factor binding, neutralising antibodies and hepatic elimination when delivered intravenously to patients. Incorporating specific targeting sequences into a cationic molecule to ensure exclusive delivery to cancer cells without the need for modifying the viral genome provides a promising and versatile new strategy for OAd engineering. Future work is aimed at developing non-toxic, virus-shielding agents for clinical translation of our highly cancer-selective OAds.

## Figures and Tables

**Figure 1 ijms-23-08884-f001:**
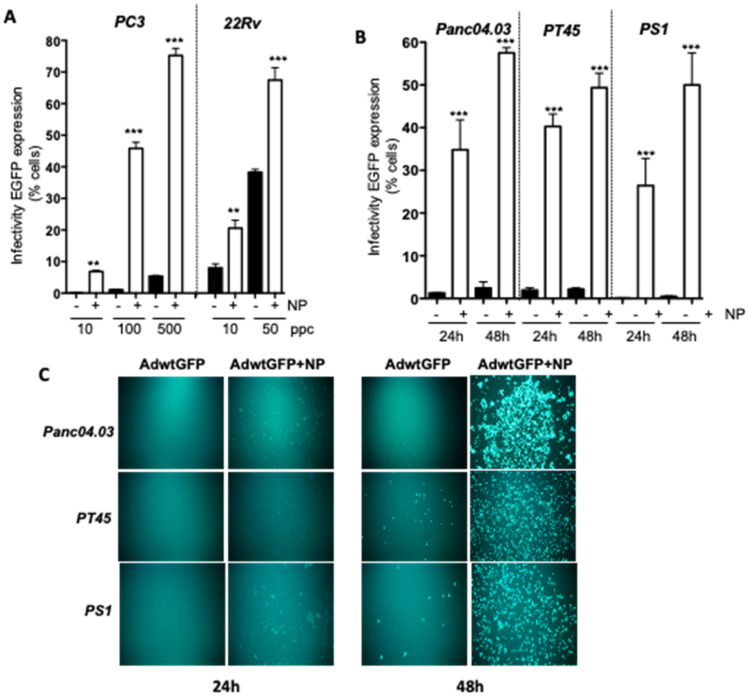
Cationic AuNP combined with Ad5wtGFP enhanced viral uptake in PCa and PDAC cell lines. (**A**) PCa cells were infected with Ad5wtGFP at 10, 100 and 500 ppc (PC3) and 10 and 50 ppc (22Rv) in the presence or absence of 0.5 pmol AuNPs. (**B**) PDAC cells were infected with Ad5wtGFP at 100 ppc and the stellate PS1 cells at 500 ppc in the presence or absence of 0.2 pmol AuNPs; samples were analysed 24 and 48 h after infection. (**A**,**B**) The EGFP signal was measured as an indication of infection levels by flow cytometry, averages ±SD, *n* = 3, ** *p* < 0.01, *** *p* < 0.001. (**C**) Fluorescent images of Panc04.03, PT45 and PS1 cells 24 and 48 h after infection with Ad5wtGFP at 50 ppc with and without AuNP at 0.2 pmol, representative of 3 studies. Images were taken at 10× magnification (Olympus IX70 microscope, Evident Europe GmbH, Stansted, UK).

**Figure 2 ijms-23-08884-f002:**
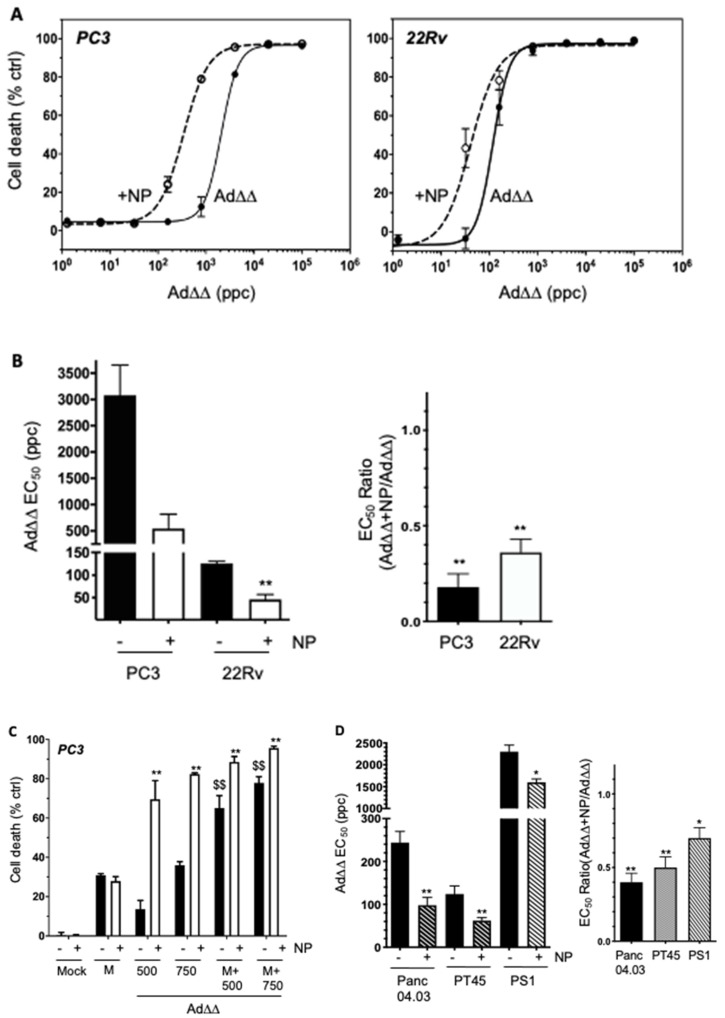
AdΔΔ-induced cell killing is enhanced by AuNPs in PCa and PDAC cells. (**A**) Dose–response to Ad∆∆ in PC3 and 22Rv cells with and without 0.1 pmol AuNPs. Cell viability determined by MTS assay. (**B**) EC_50_-values (left panel) and the relative decreases in EC_50_-values (right panel) in each cell line compared to Ad∆∆ alone. (**C**) PC3 cells treated with fixed doses of AdΔΔ at 500 ppc or 750 ppc and/or mitoxantrone (450 nM) with and without AuNPs at 0.1 pmol. Cell viability was measured by MTS assay 4d after infection, ** *p* < 0.01 (Ad/AuNP vs. Ad) and ^$$^
*p* < 0.01 (Ad/M vs. Ad). (**D**) EC_50_-values generated from Ad∆∆ dose–response curves ± AuNPs at 0.2 pmol (left panel) and the corresponding relative decreases in Panc04.03, PT45 and PS1 cells compared to Ad∆∆ alone. (**A**–**D**) Averages ± SD, *n* = 3, * *p* < 0.05, ** *p* < 0.01.

**Figure 3 ijms-23-08884-f003:**
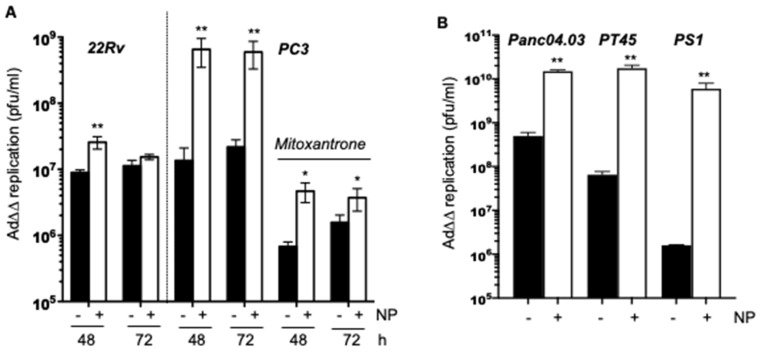
Ad∆∆ replication is enhanced in the presence of AuNP and efficiently eliminates PDAC cells in three-dimensional co-cultures with PS1 stellate cells. (**A**) PC3 and 22 Rv cells were infected with AdΔΔ at 100 ppc in the presence or absence of 0.1 pmol AuNPs. PC3 cells were also treated with mitoxantrone (450 nM) and infected with Ad∆∆ with and without preincubation with 0.5 pmol AuNP. (**B**) Panc04.03 and PT45 cells were infected at 100 pc and PS1 cells at 500 ppc with and without AuNPs at 0.2 pmol and analysed 48 h after infection. (**A**,**B**) Viral replication was determined by TCID50 assays 48 and 72 h (Pca cells) and 48 h (PDAC cells) after infection, averages ± SD, *n* = 3, * *p* < 0.05, ** *p* < 0.01. (**C**) Co-cultures of Panc04.03:PS1 (1:2) cells infected with Ad5wt (1000 or 2000 ppc) in the presence or absence of AuNP (0.5 pmol), H/E staining. Cells were cultured for 3d prior to infection with virus ± AuNPs, fixed and processed for IHC 5d post-infection, 8d-old cultures, 10× magnification, representative of three biological repeats. (**D**) Confocal images of co-cultures detailed in (**C**). Localisation of virus detected by GFP-labelled secondary antibody to the E1A-antibody (green) and nuclear DAPI stain (blue) (10× magnification).

**Figure 4 ijms-23-08884-f004:**
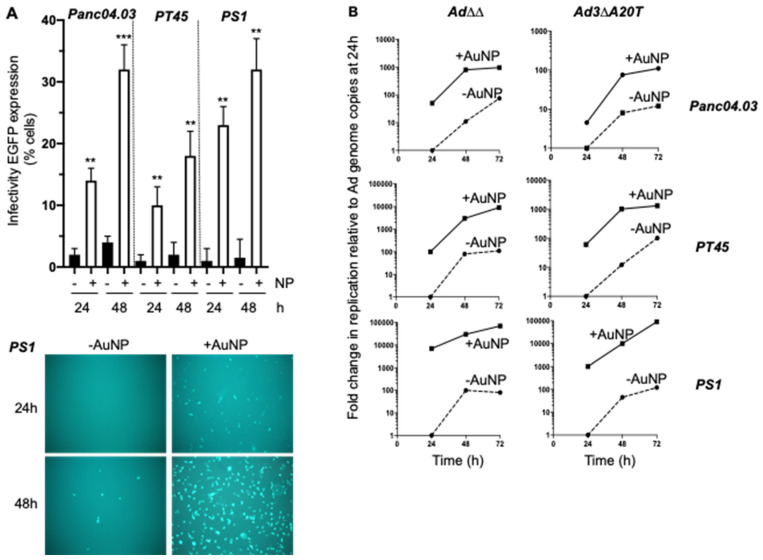
Ad-3∆-A20T infection and replication is enhanced in the presence of AuNP in PDAC cells. (**A**) Panc04.03 and PT45 were infected with Ad-3∆-A20T-GFP (100 ppc) and PS1 (500 ppc) in the presence or absence of 0.2 pmol AuNPs; samples were analysed 24 and 48 h after infection by flow cytometry for GFP expression (left panel), averages ± SD, *n* = 3, ** *p* < 0.01, *** *p* < 0.001. Lower panel: representative fluorescent images of PS1 cells 24 h and 48 h after infection as in (**A**), images were taken at 10× magnification (Olympus IX70 microscope). (**B**) Replication rate in Panc04.03, PT45 and PS1 cells, determined by qPCR for viral genome copies. Cells were infected as above with Ad∆∆ and Ad-3∆-A20T in the presence or absence of AuNPs at 0.2 pmol and viral DNA quantified after 24, 48 and 72 h. Data presented from one experiment in triplicates relative to the respective virus alone at 24 h, averages ± SEM.

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
