# Peer review of "Complexing the Oncolytic Adenoviruses Ad∆∆ and Ad-3∆-A20T with Cationic Nanoparticles Enhances Viral Infection and Spread in Prostate and Pancreatic Cancer Models"

_ijms, 2022, doi:10.3390/ijms23168884_

Round 1

Reviewer 1 Report

Comments:

This manuscript “Complexing the oncolytic adenoviruses Ad∆∆ and Ad-3∆-A20T with cationic nanoparticles enhance viral infection and spread in prostate and pancreatic cancer models” describes load and delivery of oncolytic adenoviruses using cationic gold nanoparticles for anticancer therapy. Overall, it needs a major revision to be published.

1. In introduction, please also discuss the importance of abscopal effects in addition to the systemic delivery.

2. It has been reported that cationic surface charges are less benefit than neutral and negative surface charges in terms of systemic circulation, non-specific systemic elimination although cellular uptake to cancer cells can be increased in vitro.

3. Please measure the size and zeta potential of complexation.

4. Please briefly describe the characteristics and composition of cationic AuNPs.

5. In gene delivery, the intracellular dissociation between genes and cationic gene delivery systems governs the transfection efficacy. Please describe the working mechanism of AuNP in the cells; How to decomplex after uptaking into the cells?

Author Response

Reviewer 1: Comments: This manuscript “Complexing the oncolytic adenoviruses Ad∆∆ and Ad-3∆-A20T with cationic nanoparticles enhance viral infection and spread in prostate and pancreatic cancer models” describes load and delivery of oncolytic adenoviruses using cationic gold nanoparticles for anticancer therapy. Overall, it needs a major revision to be published.

            Response: We thank  the reviewer for the feedback and hope we have satisfactory answered all below.

  1. In introduction, please also discuss the importance of abscopal effects in addition to the systemic delivery.

Response: We have now included the following new section in the  introduction. ‘When OAds were combined with immune checkpoint inhibitors (e.g. anti-PD-1) or cytokines the activation of the host immune system resulted in elimination of non-treated tumours and long-term immunity [8,9]. Another example of the abscopal effect is the spread of OAds from the tumour site to distant tumours via extracellular vesicles [10].’

  1. It has been reported that cationic surface charges are less benefit than neutral and negative surface charges in terms of systemic circulation, non-specific systemic elimination although cellular uptake to cancer cells can be increased in vitro.

Response: This is a very complex issue and the reviewer may have additional evidence of beneficial effects by neutral and negatively charged compounds in the circulation. It has been described that coating Ad vectors with PEG as well as with other neutral polymers or gold nanoparticles could efficiently shield against extracellular components but decreased Ad bioactivity. However, in our study we focused on cellular uptake in 2D- and 3D- and in PD studies in mice and found that cationic compounds associated with our adenoviral mutants significantly increased cancer cell uptake followed by replication and spread in all cell culture models and that the virus also reached and propagated in human tumor xenografts after systemic delivery. See for example [Pandi et al.,  Int. J Pharm.  Vol 550; 2018] that exemplifies that negatively charged nucleic acids are more efficiently delivered when encapsulated in lipoplexes with positive charge, also in agreement with numerous cationic transfection reagents.

  1. Please measure the size and zeta potential of complexation.

Response: In our previous work we characterized the size and charge of the AuNPs alone and the Ad-AuNPs complexes [Gonzalez-Pastor R., et al., 2021, Acta Biomater, 134; Hernandez Y., et al., 2019, RSC Advances, 9(3)]. Different studies have shown the zeta potential of naked Ad to be negative [Jung SJ, et al. Biomacromolecules 2015, 16(1); Kim SY, et al. Drug Deliv 2018, 25(1)]. We determined the zeta potential of the AuNPs, obtaining values of -25 mV (pH 4-10) before the functionalization and of +30 mV for AuNPs functionalized with ammonium groups (pH 4-10). We also determined the change in charges of the AuNPs after functionalization by electrophoretic mobility on agarose gel. In contrast to the positive charge provided by primary and secondary amines, quaternary ammonium groups contribute with a high positive charge that remains constant along the whole range of pH. Additionally, TEM images reveal that electrostatically-induced aggregation led to the formation of complexes between 150-280 nm containing several Ad capsids. Altogether, we can state that the Ad surface is successfully coated with AuNPs through electrostatic interactions and that positively-charge AuNPs shield the negative charges, resulting in Ad@AuNPs complexes with a net positive charge that increases cell uptake.

We have included this information in the materials and methods and results section.

  1. Please briefly describe the characteristics and composition of cationic AuNPs.

            Response: Round-shape gold nanoparticles of 14 nm were synthesized by reduction with  sodium citrate and further functionalized with thiolated poly (ethylene glycol) (SH-EG(8)–(CH2)2–COOH, PEG-COOH, Iris Biotech), followed by the covalent conjugation of (2-aminoethyl)trimethylammonium chloride hydrochloride (R’R3N+). UV-Vis spectroscopy absorption spectrum of AuNPs only functionalized with PEG (Au@COOH) and NR4+ (Au@NR4+) exhibited a sharp peak at 520 nm, corresponding to the localized surface plasmon    resonance band (LSPR) of 14 nm monodisperse spherical AuNPs. LSPR bands of complexes   Ad@Au@NR4+ red-shifted from 520 nm to 547 nm; in contrast, Au@COOH were not capable of binding to Ad and the LSPR band remained unchanged (Ad@Au@COOH).

  1. In gene delivery, the intracellular dissociation between genes and cationic gene delivery systems governs the transfection efficacy. Please describe the working mechanism of AuNP in  the cells; How to decomplex after uptaking into the cells?

            Response: This is a very interesting aspect of the delivery process that would require further ivestigation. From our previous studies we know that the AuNPs are taken up into the cell together with the virus [Hernandez Y., et al., 2019, RSC Advances, 9(3); Gonzalez-Pastor R., et   al., 2021, Acta Biomater, 134]. We speculate that the complexes are internalized mostly via        macropinocytosis (clusters or complexes at the same time) and via endocytosis  (individual  complexes), similar to the internalization of naked adenovirus. The acidic pH of the endosome facilitates viral uncoating but would also facilitate the dissociation of the non-covalent interactions between the positively charged AuNPs and the Ads, followed by the regular viral DNA transport to the nucleus.

Reviewer 2 Report

I appreciate the authors for the effort on manuscript entitled "Complexing the oncolytic adenoviruses Ad∆∆ and Ad-3∆-A20T with cationic nanoparticles enhance viral infection and spread in prostate and pancreatic cancer models" .

I feel authors should consider the below modifications and suggestions to improve the worth of the manuscript.

Abstract: It was written in general way, It would be great if the authors divide into aim/scope of the study, methods with results (specific in numbers) and conclusion in brief. 

Introduction: Authors mentioned about TME and also cationic nanoparticles for delivery. Kindly refer to the below articles: 

Exploring the Potential of Nanotherapeutics in Targeting Tumor Microenvironment for Cancer Therapy: https://doi.org/10.1016/j.phrs.2017.05.010

Cationic liposomes for co-delivery of paclitaxel and anti-Plk1 siRNA to achieve enhanced efficacy in breast cancer: https://doi.org/10.1016/j.jddst.2018.09.017

Comparison of cationic liposome and PAMAM dendrimer for delivery of anti-Plk1 siRNA in breast cancer treatment: https://doi.org/10.1080/10837450.2019.1567763

Dendrimer as a new potential carrier for topical delivery of siRNA: A comparative study of dendriplex vs. lipoplex for delivery of TNF-α siRNA: https://doi.org/10.1016/j.ijpharm.2018.08.024

Results: Cationic AuNP combined with Ad5wtGFP enhanced viral uptake in PCa and PDAC cell lines: What is the reason behind the selection of these two cell lines specifically?

Authors performed uptake in cells with 10 X magnification, it would be great if the authors might have taken in higher magnification or with Confocal imaging.

 AdΔΔ-induced cell killing is enhanced by AuNPs in PCa and PDAC cells: It would be great authors could include control in the groups.

What is significance of modification of surface charge of the particles on uptake? It would be good to include in the discussion section.

English editing is required to improve the quality of manuscript.

Author Response

Reviewer 2: I appreciate the authors for the effort on manuscript entitled "Complexing the oncolytic adenoviruses Ad∆∆ and Ad-3∆-A20T with cationic nanoparticles enhance viral infection and spread in prostate and pancreatic cancer models".  I feel authors should consider the below modifications and suggestions to improve the worth of the manuscript

            Response: We thank the reviewer for the thoughtful feedback on the current manuscript and hope we have satisfactory answered all queries below

  1. Abstract: It was written in general way, It would be great if the authors divide into aim/scope of the study, methods with results (specific in numbers) and conclusion in brief.

            Response: We have now modified the abstract according to the reviewer’s suggestion:

Oncolytic adenoviruses (OAd) can be employed to efficiently eliminate cancer cells through multiple mechanisms of action including cell lysis and immune activation. Our OAds AdΔΔ and Ad-3∆-A20T selectively infect, replicate in and kill adenocarcinoma cells, and re-sensitise drug-resistant cells in preclinical models. Further modifications are required to enable systemic delivery in patients due to the rapid hepatic elimination and neutralisation by blood factors and antibodies. Here, we show data that support the use of coating OAds with gold nanoparticles (AuNPs) as a possible new method of virus modification to help augment tumour uptake. Pre-incubation of cationic AuNPs with AdΔΔ, Ad-3∆-A20T and wild type adenovirus (Ad5wt) was performed prior to infection of prostate/pancreatic cancer cell lines (22Rv, PC3, Panc04.03, PT45) and a pancreatic stellate cell line (PS1). Levels of viral infection, replication and cell viability were quantified 24-72h post infection in the presence and absence of AuNPs. Viral spread was assessed in organotypic cultures. Uptake of Ad∆∆, Ad-3∆-A20T and Ad5wt was significantly increased in the presence of AuNPs in all cell lines tested (ranging from 1.5-fold to 40-fold) compared to virus alone with the greatest uptake observed in PS1, a usually adenovirus-resistant cell line. Corroborating with this, the presence of AuNP increased viral replication of AdΔΔ and Ad-3∆-A20T in all cell lines leading to increased cell killing with the greatest increases in the most virus insensitive cells (from 1.4-fold to 5-fold). To conclude, electrostatic association of virus with cationic agents provides a new avenue to increase the viral dose in tumour lesions and a promising approach to protect the virus from detrimental blood factor binding that warrants further investigation for clinical translation.

  1. Introduction: Authors mentioned about TME and also cationic nanoparticles for delivery. Kindly refer to the below articles: 

Exploring the Potential of Nanotherapeutics in Targeting Tumor Microenvironment for Cancer Therapy: https://doi.org/10.1016/j.phrs.2017.05.010

Cationic liposomes for co-delivery of paclitaxel and anti-Plk1 siRNA to achieve enhanced efficacy in breast cancer: https://doi.org/10.1016/j.jddst.2018.09.017

Comparison of cationic liposome and PAMAM dendrimer for delivery of anti-Plk1 siRNA in breast cancer treatment: https://doi.org/10.1080/10837450.2019.1567763

Dendrimer as a new potential carrier for topical delivery of siRNA: A comparative study of dendriplex vs. lipoplex for delivery of TNF-α siRNA: https://doi.org/10.1016/j.ijpharm.2018.08.024

            Response: We thank the reviewer for the suggested literature. The publications are very interesting and informative however, we did not design our studies to specifically target the TME. Furthermore the dendrimers and liposomes used in the  publications were developed to deliver  small molecules or siRNA and are not suitable for adenovirus delivery. PEG-PEI and liposomes      have been explored for adenovirus delivery but were found to greatly attenuate or prevent viral cancer cell entry. We mention the TME in the introduction to highlight the well-established fact that the stroma, associated with PDAC is a major obstacle for efficient therapy. Our AuNPs may shield the virus from detrimental blood factors so that a higher dose will reach the tumour sites after systemic delivery. Once at the tumour site the virus-AuNP will attach either directly to cancer cells or to stromal cells followed by internalisation and viral propagation can proceed only in  cells with deregulated cell cycle control. Our organotypic cultures and the proof-of-concept in   vivo study demonstrate that virus is indeed infecting both cancer-associated stromal and cancer cells enabling viral replication in cells with altered cell cycle regulation. 

  1. Results: Cationic AuNP combined with Ad5wtGFP enhanced viral uptake in PCa and PDAC cell lines: What is the reason behind the selection of these two cell lines specifically?

Response: Both PCa and PDAC are cancers of high unmet medical need. The survival rate is relatively long for PCa (5-15 years) and very short for PDAC (6-12months) however, in the final  stages there is no cure for either cancer and patients suffer from severe pain and morbidity. Our team is focused on developing novel therapeutic strategies to benefit these patients (addressed in the first paragraph of the introduction). The human cell lines used in the study have the typical genetic alterations identified in PCa (PC3 represent metastatic androgen-independent PCa and 22Rv1 represent an earlier stage of PCa and express wild type and mutant AR; the PT45 and Panc04.03 express the mutant and constitutively active KRas typical of PDAC cancer cells).

  1. Authors performed uptake in cells with 10 X magnification, it would be great if the authors might have taken in higher magnification or with Confocal imaging.

Response: We are unsure which images the reviewer is referring to but perhaps the fluorescent images in Fig. 1C and 4A (?) of cells grown on plastic illustrating that more cells are infected with virus in the presence of AuNPs. We chose the 10x magnification to better demonstrate this as a higher magnification would only result in a smaller field of view with fewer cells. Under these culture conditions confocal microscopy was not an option. Also note that the virus only expresses GFP when internalised and the genome is transported to the nucleus for mRNA translation and consequently the GFP signal is only present in cells infected with active virus. In Fig. 3D we show infection in organotypic cultures using confocal microscopy and in Supplementary  Fig. 4., the organotypics are shown at 20x magnification.

  1. AdΔΔ-induced cell killing is enhanced by AuNPs in PCa and PDAC cells: It would be great authors could include control in the groups.

Response: We have included the following controls in the cell killing assays demonstrated in  Fig. 2; virus alone (A), EC50 values with and without AuNPs (B and D), AuNPs alone, Mitoxantrone alone, virus alone, Mitoxantrone + AuNPs (C). Additional AuNPs alone are shown in Supplementary  Fig 1, and Supplementary Fig. 3 and 4. 

  1. What is significance of modification of surface charge of the particles on uptake? It would be good to include in the discussion section.

Response: As indicated by the reviewer, this is a critical aspect affecting the transduction efficiency and we focused on this issue in our previous work [Y. Hernandez, et al. RSC Advances, 2019, 9(3); Gonzalez-Pastor R., et al., 2021, Acta Biomater, 134]. The improved uptake of virus in cancer cells is completely dependent on the positively charge of the AuNPs. Adenovirus is overall negatively charged and infects by binding to cellular receptors (CAR and integrins). By complexing a cationic molecule (AuNPs or proteins etc.) with virus, cells with low level receptor expression can be infected at a higher level because of the positively charges that can associate with the negative cell membrane and bring the virus closer to the receptors, acting as a bridging agent. In the first work, we tested the impact of the positive charge by comparing the cell transduction levels of Ad coated with Au@NR4+ and Au@RGD-NR4+ (the AuNPs used in the current work) and with Au@RGD (AuNPs functionalized with RGD but without the quaternary ammonium groups). In all our previous experiments, we observed how the presence of the quaternary ammonium groups even doubled the transgene expression levels compared to Ads coated with AuNPs without quaternary ammonium groups. In the second work, we confirmed that those AuNPs only functionalized with thiolated PEG and with no quaternary ammonium groups did not bind to the Ads and did not promote transduction. Additionally, we compared the transduction efficiency of uncoated and coated Ad with our AuNPs in a panel of cell lines with different expression levels of CAR and integrins, and concluded that there was an increase in transduction regardless of their level of CAR expression, that can be mainly attributed to a more efficient cell internalization mediated by electrostatic interactions, although a minor contribution of the RGD-αvβ5 interactions on the internalization process was suggested.  

This information is included in the discussion (lines 310-315 and lines 322-228).

  1. English editing is required to improve the quality of manuscript.

Response: We have now further checked and corrected grammar and spelling

Round 2

Reviewer 1 Report

The author responded successfully to the reviewer's comments.

Author Response

We thank the reviewer for appreciating our efforts to correct and improve the manuscript. 

Reviewer 2 Report

Authors does not include the adequate discussion about the results and need to be cited with relevant supporting literature.

Figure 1C, Figure 3C and Figure 4 images quality can be improved.

Authors also published on "The Novel Oncolytic Adenoviral Mutant Ad5-3D-A20T Retargeted to avb6 Integrins Efficiently Eliminates Pancreatic Cancer Cells" 

What is the Novelty of this study as compared to the one which already published?

Authors also mentioned about " complexing the viral particle with cationic molecules or agents enables a practical and versatile method for augmenting the efficacy"

Authors did not specify the toxicity and immunological complications related to the particles they used. It would be great if they can enlighten on this portion of the study.

Author Response

Reviewer 2: Authors does not include the adequate discussion about the results and need to be cited with relevant supporting literature.

Response: Throughout the discussion we are discussing the results point by point. In the current version we have included additional comments to clarify the results/conclusions with the appropriate references. Please see ‘track comments’ in manuscript.

Reviewer: Figure 1C, Figure 3C and Figure 4 images quality can be improved.

Response: We previously addressed this issue in a query by one of the reviewers and are not sure what the reviewer would want us to do. The images we submitted correctly represent the data and are of regular standard. We do not intend to manipulate these images to enhance our results. Once the manuscript is accepted, we can work with the editors to make any adjustments deemed necessary but still represents the true data.

Reviewer: Authors also published on "The Novel Oncolytic Adenoviral Mutant Ad5-3D-A20T Retargeted to avb6 Integrins Efficiently Eliminates Pancreatic Cancer Cells" 

What is the Novelty of this study as compared to the one which already published?

Response: This was described in the introduction and addressed in the discussion throughout the manuscript. Briefly, our previous publication reports on the engineering and development of the oncolytic virus Ad5-3D-A20T while the current manuscript reports on further enhancement of infectivity of several oncolytic adenoviruses with negative surface charge.

See second and third paragraph in introduction and throughout the discussion including the added statement ‘[49]. Unlike our tropism modified FMDV-targeting Ad-3∆-A20T, AuNP coating allows a more versatile cell line-independent enhancement of OAds infection.’ on line 460-462.

Reviewer: Authors also mentioned about " complexing the viral particle with cationic molecules or agents enables a practical and versatile method for augmenting the efficacy"

Authors did not specify the toxicity and immunological complications related to the particles they used. It would be great if they can enlighten on this portion of the study.

Response:  We have now made the previous text in the discussion section more clear to address the  reviewer’s comment, see lines 463-469: ‘Although gold particles may accumulate in the liver to toxic levels, we previously demonstrated insignificant hepatotoxicity and no significant differences in the number of white blood cells when Ad5/AuNP complexes or AuNPs alone were administered systemically in murine models at high doses [36]. Further development of other non-toxic cationic compounds as alternative OAd-shielding agents could support clinical translation.’

Round 3

Reviewer 2 Report

I Appreciate authors for revision. Overall, the manuscript looks comparatively better. I feel the authors should check on minor grammatical errors in the manuscript. 

Author Response

We appreciate the reviewer’s careful reading of our manuscript and have now edited the  language accordingly. One of the authors is a native English speaker (Dr Man) who reviewed and altered the text significantly to British standard language, see ‘track changes’ in manuscript. We hope that this improved manuscript is now acceptable for publication.

This manuscript is a resubmission of an earlier submission. The following is a list of the peer review reports and author responses from that submission.